# Learning a neural response metric for retinal prosthesis

Nishal P. Shah[1, 2], Sasidhar Madugula[1], E.J. Chichilnisky[1], Yoram Singer[2, 3], and Jonathon Shlens[2]

[1]Stanford University
[2]Google Brain
[3]Princeton University

## Abstract

Retinal prostheses for treating incurable blindness are designed to electrically stimulate surviving retinal neurons, causing them to send artificial visual signals to the brain. However, electrical stimulation generally cannot precisely reproduce typical patterns of neural activity in the retina. Therefore, an electrical stimulus must be selected so as to produce a neural response as close as possible to the desired response. This requires a technique for computing the distance between a desired response and an achievable response that is meaningful in terms of the visual signal being conveyed. We propose a method to learn a metric on neural responses directly from recorded light responses of a population of retinal ganglion cells (RGCs) in the primate retina. The learned metric produces a measure of similarity of RGC population responses that accurately reflects the similarity of visual inputs. Using data from electrical stimulation experiments, we demonstrate that the learned metric could produce improvements in the performance of a retinal prosthesis.

## 1 Introduction

An important application of neuroscience research is the development of electronic devices to replace the function of diseased or damaged neural circuits (Wilson et al., 1991; Schwartz, 2004). Artificial vision has been a particularly challenging modality due to the richness of visual information, its diverse uses in perception and behavior, and the complexity of fabricating a device that can interface effectively with neural circuitry (Stingl et al., 2013; Wilke et al., 2011; Jepson et al., 2014a).

The most advanced example is a retinal prosthesis: a device that replaces the function of neural circuitry in the retina lost to degenerative disease. Most of the computational work related to this application has focused on building encoding models that use the visual image to accurately predict the spiking activity of populations of retinal ganglion cells (RGCs), the output neurons of the retina that convey visual information to the brain. Leading models include linear models (Chichilnisky, 2001), probabilistic point-process models (Pillow et al., 2008) and recently proposed models employing rich nonlinearities (McIntosh et al.; Batty et al.; Shah et al., 2017).

However, an accurate encoding model, although valuable, is insufficient. Any retinal prosthesis – whether based on electrical stimulation (Sekirnjak et al., 2008) or optical stimulation (Boyden et al., 2005; Bernstein et al., 2008) – is limited in its ability to create arbitrary desired patterns of neural activity, due to inefficiencies or lack of specificity in the stimulation modality (Barrett et al., 2014; Jepson et al., 2014a). Thus, a given stimulation system can only achieve a limited vocabulary of elicited spike patterns. Although a powerful and accurate encoding model might indicate that a particular spike pattern would be the natural biological response to the incident visual stimulus, the desired spike pattern might not reside within the feasible set of the stimulation device (Figure 1).

Previous studies (Jepson et al., 2014b) have addressed this problem by selecting the electrical stimulation which minimizes the number of unmatched spikes across cells – equivalent to the Hamming distance between two binary vectors. Even though a Hamming distance is easy to compute, this solution is not necessarily optimal. The goal of a prosthetics device should be to instead select an

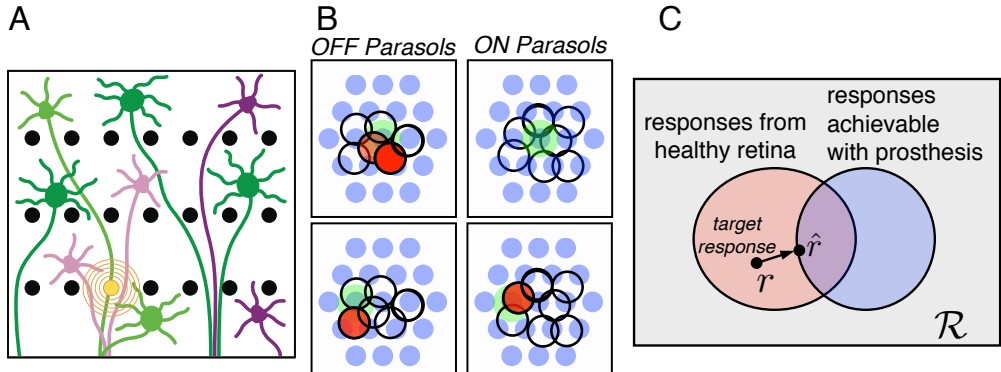

Figure 1: Setup of a retinal prosthetic system. A. Cartoon showing ON (purple) and OFF(green) RGCs overlayed on top of an electrode array (black dots). The cells firing due to passing current through the orange electrode are lightly shaded. Note that a cell can fire even if an axon of the cell passes near the stimulating electrode. Hence, single electrode stimulation leads to firing of many cells. B. *Real* recorded spiking activity in a two populations of primate retinal ganglion cells (RGCs) demonstrating the lack of specificity from electrical stimulation. The electrodes are in blue and the stimulated electrode is shaded green. C. The target firing pattern $r$ often lies outside the set of firing patterns achievable with the prosthesis. The goal of the learned metric is to define a distance measure to identify the nearest feasible electrical stimulation $\hat{r}$. $\mathcal{R}$ denotes the set of all neural population responses.

electrical stimulation pattern that produces a response as close as possible to the desired pattern of activity in terms of the elicited visual sensation (Figure 1C). In lieu of measuring the visual sensation produced by a prosthetic, we instead posit that one may infer a distance metric based on the signal and noise properties of individual and populations of neurons (Shlens et al., 2009; Pillow et al., 2008; Field & Chichilnisky, 2007). In contrast, previous approaches to spike metrics have focused on user-specified, parameteric functions (Victor & Purpura, 1996; 1997; Victor, 2005) or unsupervised techniques to cluster nearby spike patterns (van Rossum, 2001; Dubbs et al., 2010; Ganmor et al., 2015).

In this work, we propose a neural response metric learned directly from the statistics and structure of firing patterns in neural populations, with the aim of using it to select optimal electrical stimulation patterns in a prosthesis device. In particular, we learn a neural response metric by applying ideas from metric learning to recordings of RGC populations in non-human primate retina. We demonstrate that the learned metric provides an intuitive, meaningful representation of the similarity between spike patterns in the RGC population, capturing the statistics of neural responses as well as similarity between visual images. Finally, we use this metric to select the optimal electrical stimulation pattern within the constraints of the electrical interface to a population of RGCs.

## 2 METRIC AND SIMILARITY LEARNING

In this section we describe the algorithmic framework for learning pseudometrics or similarity measures in neural response space. We start by introducing notations and conventions that we use throughout the paper. We use bold face letters to denote vectors and upper case letters to denote matrices. We denote the symmetrization operator of a square matrix $M$ by $\text{sym}(M) = \frac{1}{2}(M + M^{\top})$.

A single frame of visual stimulus, $\mathbf{s}$, is an image represented as an $n \times n$ matrix. The space of possible stimuli is $\mathcal{S} \subset \mathbb{R}^{n \times n}$. A sequence $\mathbf{s}_l, \ldots, \mathbf{s}_m$ of $m - l + 1$ frames, where $\mathbf{s}_j \in \mathcal{S}$, is denoted as $\mathbf{s}_{l:m}$. In order to simplify our notation, we define the responses of the cells to be a $p$ dimensional vector and the space of possible responses as $\mathbf{r} \subseteq \mathbb{R}^p$. Analogously, a sequence of cell activities $\mathbf{r}_t$ for $t = l, \ldots, m$ is denoted $\mathbf{r}_{l:m}$. To simplify the presentation below, we confine the visual stimulus to be a single image and the corresponding response of each cell to be a scalar.

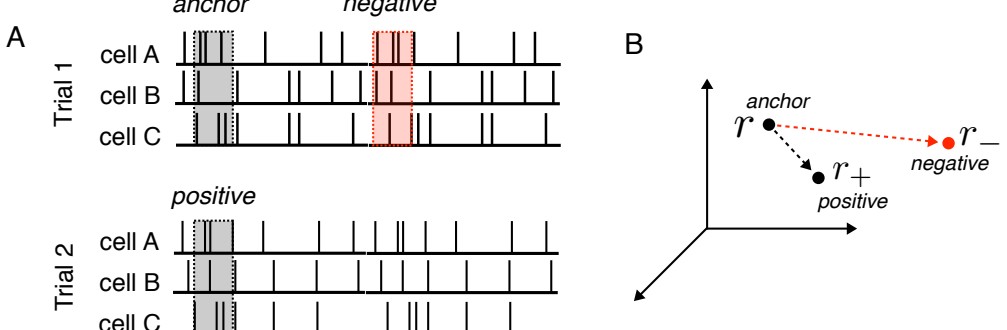

Figure 2: Method for constructing an objective for learning a neural metric using triplet-based losses. (A) Schematic of spike trains from population of neurons responding to a dynamic stimulus (not shown). The spiking response across the window is randomly selected at relative time $t$. The binarized representation for the population response is termed the *anchor*. In a second presentation of the same stimulus, the population response at the same relative time $t$ is recorded and labeled a *positive*. The *negative* response is any other selected time. (B) The objective of the triplet-loss is to require that the *anchor* is closer to the *positive* response than the *negative* response with a margin. See text for details.

## 2.1 RELATED WORK

Metric and similarity learning using constrative loss (Chopra et al., 2005; Hadsell et al., 2006) and triplets loss (Shalev-Shwartz et al., 2004; Weinberger & Saul, 2009) have been used extensively in several domains. In computer vision, these methods achieve state-of-the-art performance on face recognition (Schroff et al., 2015; Sun et al., 2014; Taigman et al., 2014) and image retrieval (Wang et al., 2014). A central theme of this work has focused on improving metric learning by mining semi-hard negatives (Schroff et al., 2015). Because many negatives provide minimal information, these methods use a partially learned metric to identify negatives that may maximally improve the quality of the metric given a fixed number of updates. To avoid the computational burden imposed by such methods, some works have proposed alternative loss functions which either make efficient use of all the negatives in a batch (Oh Song et al., 2016) or multiple classes using n-tuplets (Sohn, 2016). Our method is similar to these methods as we make efficient use of all the negatives in a batch as in (Oh Song et al., 2016) but also use a simplified, softmax-based loss function (Sohn, 2016).

## 2.2 EMPIRICAL LOSS MINIMIZATION

Given the population response space $\mathcal{R}$, we learn a function $h : \mathcal{R} \times \mathcal{R} \to \mathbb{R}$ which captures invariances in the spiking response when the same stimulus is presented multiple times. The scoring function $h$ is viewed either as a similarity function or a pseudometric. To distinguish between the two cases, we use $d(\cdot, \cdot)$ to denote a pseudometric. A pseudometric $d$ needs to satisfy:

**Positivity.** $d(\mathbf{r}_1, \mathbf{r}_2) \geq 0$ and $d(\mathbf{r}, \mathbf{r}) = 0$

**Sub-additivity.** $d(\mathbf{r}_1, \mathbf{r}_2) + d(\mathbf{r}_2, \mathbf{r}_3) \geq d(\mathbf{r}_1, \mathbf{r}_3)$

**Symmetry.** $d(\mathbf{r}_1, \mathbf{r}_2) = d(\mathbf{r}_2, \mathbf{r}_1)$

During the experiments, repeats of the same sequence of visual stimuli are presented. The responses collected during the $i$th presentation (repeat) of visual stimulus are denoted by $(\mathbf{s}_t^i, \mathbf{r}_t^i)$. Here $\mathbf{s}_t^i$ is the stimulus history which elicits population response $\mathbf{r}_t^i$ at time $t$. The goal of this approach is to learn a metric such that pairs of responses generated during different repeats of the same stimulus are closer, or more similar, than pairs of responses generated by different stimuli. We slice the data into triplets of the form $(\mathbf{r}, \mathbf{r}_+, \mathbf{r}_-)$ where $\mathbf{r}$ and $\mathbf{r}_+$ are responses of cells to the same stimulus while $\mathbf{r}_-$ is the response to a different visual stimuli (Figure 2A). We refer to $(\mathbf{r}, \mathbf{r}_+)$ as a positive pair and $(\mathbf{r}, \mathbf{r}_-)$ as a negative pair (Figure 2B).

A common method to improve the learned metrics is to choose difficult negatives as described above. As it can be computationally demanding to mine hard negatives, we found that a much simpler

strategy of randomly sampling a common set of negatives for all the positive examples in the batch is effective. Hence we first sample positive pairs of responses corresponding to random stimulus times and a common set of negative responses generated by stimuli distinct from any stimulus for positive responses. Hence a batch of triplets is denoted by $\mathcal{T} = \{\{\mathbf{r}^i, \mathbf{r}^i_+\}, \{\mathbf{r}^j_-\}\}$.

Given a training set of triplets $\mathcal{T}$, the goal is to find a pseudometric such that for most $(\mathbf{r}^i, \mathbf{r}^i_+, \{\mathbf{r}^j_-\}) \in \mathcal{T}$ the distance between responses of two repeats of same stimulus is smaller than their distance to any of the irrelevant response vectors,

$$d(\mathbf{r}^i, \mathbf{r}^i_+) < \min_j d(\mathbf{r}^i, \mathbf{r}^j_-) \tag{1}$$

We cast the learning task as empirical risk minimization of the form,

$$\frac{1}{|\mathcal{T}|} \sum_{(\mathbf{r}^i, \mathbf{r}^i_+, \{\mathbf{r}^j_-\}) \in \mathcal{T}} \ell(\mathbf{r}^i, \mathbf{r}^i_+, \{\mathbf{r}^j_-\}) \,,$$

where $\ell()$ is a differential, typically convex, relaxation of the ordering constraints from (1). We use the following,

$$\ell(\mathbf{r}^i, \mathbf{r}^i_+, \{\mathbf{r}^j_-\}) = \beta \log \left[ 1 + \sum_j e^{\frac{d(\mathbf{r}^i, \mathbf{r}^i_+) - d(\mathbf{r}^i, \mathbf{r}^j_-)}{\beta}} \right] \,,$$

as the surrogate loss. We set $\beta = 10$ in our implementation.

In the case of similarity learning, we swap the role of the pairs and define,

$$\ell(\mathbf{r}^i, \mathbf{r}^i_+, \{\mathbf{r}^j_-\}) = \beta \log \left[ 1 + \sum_j e^{\frac{h(\mathbf{r}^i, \mathbf{r}^j_-) - h(\mathbf{r}^i, \mathbf{r}^i_+)}{\beta}} \right] \,,$$

We implemented two parametric forms for distance and similarity functions. The first is a quadratic form where $A \succeq 0$ and

$$h_A(\mathbf{r}_1, \mathbf{r}_2) = \mathbf{r}_1^\top A \, \mathbf{r}_2 \quad \text{and} \quad d_A(\mathbf{r}_1, \mathbf{r}_2) = (\mathbf{r}_1 - \mathbf{r}_2)^\top A \, (\mathbf{r}_1 - \mathbf{r}_2) \,. \tag{2}$$

We learn the parameters by minimizing the loss using Adagrad (Duchi et al., 2011). We project $A$ onto the space of positive semi-definite matrices space after every update using singular value decomposition. Concretely, we rewrite $A$ as, $UDU^\top$ where $U$ is a unitary matrix and $D$ is a diagonal matrix. We then threshold the diagonal elements of $D$ to be non-negative.

## 2.3 Extending metric spaces for unobserved neural populations

The quadratic metric provides a good demonstration of the hypothesis that a learned metric space may be suitable. However, a quadratic metric is not feasible for a real prosthetic device because such a metric must be trained on visually-evoked spiking activity of a neural population. In a retinal prosthetic, such data are not available because the retina does not respond to light. Furthermore, a quadratic model contains limited modeling capacity to capture nonlinear visual processing in the retina (Field & Chichilnisky, 2007).

To address these issues, we introduce a nonlinear embedding based on a convolutional neural network (CNN). The CNN encodes each cell's spiking responses in an embedding space grouped by cell type and cell body location before performing a series of nonlinear operations to map the response embedding from the response space $\mathcal{R}$ to $\mathbb{R}^p$. One benefit of this approach is that this model has an embedding dimensionality independent of the number of cells recorded while only employing knowledge of the cell body location and cell type. The cell body location and cell type are identifiable from recordings of non-visually-evoked (spontaneous) neural activity in the retina (Li et al., 2015; Richard et al., 2015).

The resulting response metric may be generalized to blind retinas by merely providing cell center and cell type information. That is, no visually-evoked spiking activity is necessary to train an embedding for a new retina. Even though the model may be fit on non visually-evoked spiking activity, this

model class is superior then the quadratic model when fit to a given retina. We discuss preliminary experiments for predicting the activity in unobserved retinas in the Discussion.

We reserve a complete discussion of model architecture and training procedure for the Appendix. In brief, we employ a hierarchical, convolutional network topology to mirror the translation invariance expected in the receptive field organization of the retina. The convolutional network consists of 595K parameters across 7 layers and employs batch normalization to accelerate training. Let $\phi(\mathbf{r})$ be the convolutional embedding of responses. The similarity and metric learned using the convolutional network is given as -

$$h_\phi(\mathbf{r}_1, \mathbf{r}_2) = \phi(\mathbf{r}_1) \cdot \phi(\mathbf{r}_2) \quad \text{and} \quad d_\phi(\mathbf{r}_1, \mathbf{r}_2) = \|\phi(\mathbf{r}_1) - \phi(\mathbf{r}_2)\|^2 . \tag{3}$$

We learn the parameters by minimizing the loss using Adam (Kingma & Ba, 2014).

## 3 RESULTS

### 3.1 EXPERIMENTAL SETUP

Spiking responses from hundreds of retinal ganglion cells (RGCs) in primate retina were recorded using a 512 electrode array system (Litke et al., 2004; Frechette et al., 2005). ON and OFF parasol RGC types were identified using visual stimulation with binary white noise and reverse correlation (Chichilnisky, 2001).

Since each analysis requires different stimulus conditions and numbers of cells, we leave the details of each preparation to the subsequent sections. For each analysis, spike trains were discretized at the 120 Hz frame rate of the display (bins of 8.33ms), and responses across all the cells for 1 time bin were used to generate each training example.

In the following sections, we quantitatively assess the quality of multiple learned metrics – each metric with increasing complexity – with respect to a baseline (Hamming distance). First, we assess the quality of the learned metrics with respect to traditional error analysis. Second, we assess the quality of the learned embeddings with respect to optimal decoding of the stimulus. Finally, we demonstrate the utility of the learned metric by employing the metric in a real electrical stimulation experiment.

### 3.2 QUANTITATIVE EVALUATION OF LEARNED METRIC SPACE.

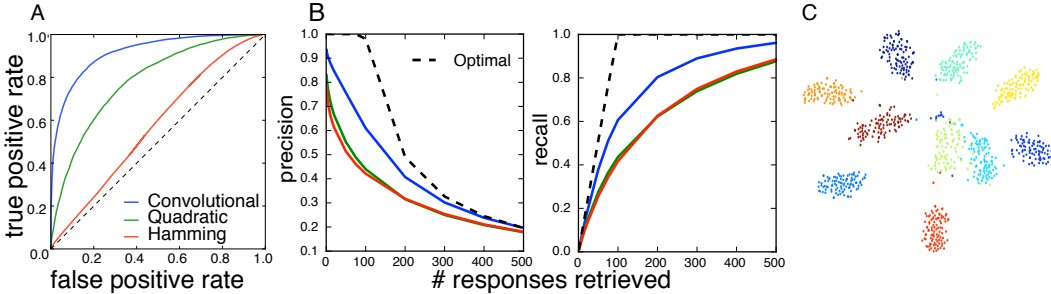

Figure 3: Quantitative analysis of learned metric. (A) ROC curve measuring false positive rate versus true positive rate across a range of thresholds for the convolutional neural network learned metric (blue, Area Under Curve (AUC:0.9), quadratic learned metric (green, AUC:0.77), and Hamming distance (red, AUC:0.56). Note that curves closer to the upper left corner correspond to higher quality metrics. (B) Precision and recall of classification as spike pattern from same stimulus or a different stimulus across increasing number of retrieved nearest neighbors from a collection of responses consisting of all 99 repeats of 10 different stimuli. The best achievable performance is shown with dashed lines. Models and colors same as (A). (C) t-SNE embedding of the collection of responses in (B) using distances estimated by the convolutional neural network. Each dot is a population response and responses with same color were generated by the same visual stimulus.

The quality of a metric in our context can be measured by its effectiveness for determining whether a pair of firing patterns arose from the same visual stimulus or from distinct visual stimuli. To evaluate the metric at the scale of large RGC populations, we focus our analysis on responses of a collection of 36 OFF parasol cells and 30 ON parasol cells to 99 repeats of a 10 second long white noise stimulus clip. The responses were partitioned into training (first 8 seconds) and testing (last 2 seconds) of each trial.

We assessed a range of learned embedding models and baselines by employing receiver-operating characteristic (ROC) analysis. Specifically, we selected the population firing pattern, $\mathbf{r}$, at a particular offset in time in the experiment (corresponding to a visual stimulus history) and compared this firing pattern to two other types of firing patterns: (1) the firing pattern from the same group of cells at the same time during a second repeated presentation of the stimulus, $\mathbf{r}_+$; and (2) the firing pattern at a distinct, randomly selected time point, $\mathbf{r}_-$. For a given threshold, if the metric results in a correct classification of $\mathbf{r}_+$ as the same stimulus, we termed the result a *true positive*. For the same threshold, if an embedding metric incorrectly classified $\mathbf{r}_-$ as the same stimulus, we termed it a *false positive*. Note that unlike training, we do not choose a common set of negatives for testing.

Figure 3A traces out the trade-off between the false positive rate and true positive rate across a range of thresholds in an assortment of embedding models for neural population activity. Better models trace out curves that bend to the upper-left of the figure. The line of equality indicates a model that is performing at chance. A simple baseline model of a Hamming distance (red curve) performs least accurately. A quadratic metric which permits variable weight for each neuron and interaction between pairs of neurons improves the performance further (green curve). Finally, replacing a quadratic metric with a euclidean distance between embedding of responses using a convolutional neural network improves the performance further (blue curve).

The ROC analysis provides strong evidence that increasingly sophisticated embedding models learn global structure above and beyond a Hamming distance metric. We also examined how the local structure of the space is captured by the embedding metric by calculating the learned embeddings on a test dataset consisting of 99 repeats each of the 10 different visual stimuli. We randomly selected a firing pattern $\mathbf{r}$ from one presentation of the stimulus, and identified $k$ nearest neighbors according to our metric, for increasing $k$. Among the $k$ nearest neighbors, we assessed precision, i.e. what fraction of the nearest neighbors correspond to 98 other presentations of the same stimulus. A perfect learned embedding model would achieve a precision of 1 for $k \leq 98$ and $98/k$ otherwise (Figure 3B, dashed). We also measured recall, i.e. what fraction of the remaining 98 presentations of the same stimulus are within the $k$ nearest neighbors. A perfect learned embedding model would achieve recall of $k/98$ for $k \leq 98$ and 1 otherwise (Figure 3B, dashed). Figure 3B highlights the performance of various learned methods across increasing $k$. The results indicate that the precision and recall are below an optimal embedding, but the convolutional metric performs better than quadratic and Hamming metrics.

To visualize the discriminability of the response metric, we embed the 99 responses to 10 distinct stimuli using t-SNE (Maaten & Hinton, 2008) with distances estimated using the convolutional metric. We see in Figure 3C that responses corresponding to same visual stimulus (same color) cluster in the same region of embedding space reflecting the ability of the response space metric to discriminate distinct stimuli.

## 3.3 LEARNED METRIC CAPTURES STIMULUS SIMILARITY.

Although we trained the metric only based on whether pairs of responses are generated by the same stimulus, Figure 3C suggests that the learned response metric provides additional discriminative stimulus information. In the following sections, we attempt to quantitatively measure how well the response metric captures stimulus information by performing stimulus reconstruction. Our hypothesis is that stimulus reconstruction provides a proxy for the ultimate goal of assessing perceptual similarity.

Stimulus reconstruction has a rich history in the neural coding literature and presents significant technical challenges. To simplify the problem, we focus on linear reconstruction (Bialek et al., 1991; Rieke et al.; Roddey & Jacobs, 1996) because the objective is clear, the problem is convex and the resulting reconstruction is information rich (Stanley et al., 1999; Berry et al., 1997). One limitation of this approach is that linear reconstruction does not capture rich nonlinearities potentially present in encoding. For this reason, we focus subsequent analysis on the quadratic and Hamming metrics

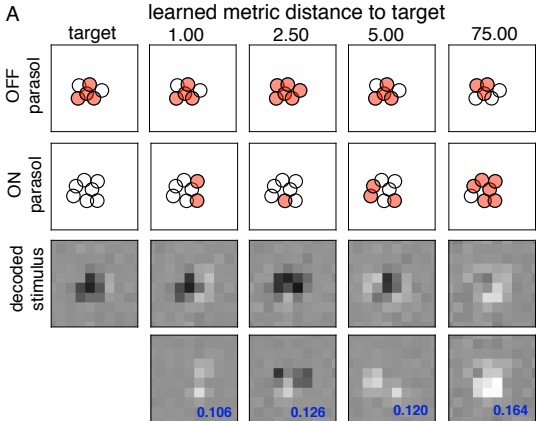
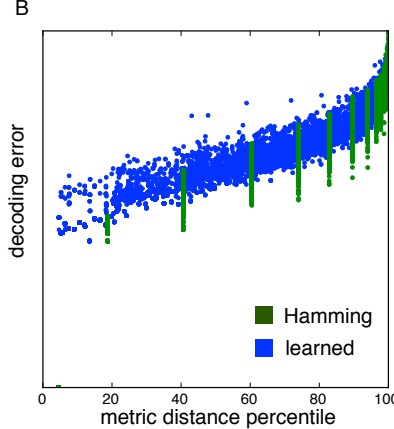

Figure 4: Decoded stimulus degrades with increasing response distance. (A) Target and retrieved responses at different distances according to a learned quadratic metric. Receptive field locations of OFF and ON parasol cells are shown (shading indicates firing). Numbers above indicate percentile of cumulative distribution across distances in learned metric space. Linearly decoded stimulus and the difference from the decoded stimulus for the target firing pattern is shown below, respectively. Numbers in the plot (blue) indicate MSE between decoded stimulus for target and that for the firing pattern above. (B) Hamming distance (green) and learned quadratic metric distance (blue) between a randomly selected pair of firing patterns plotted versus the MSE between corresponding linearly decoded stimuli.

and reserve the analysis of the nonlinear embedding for future work with nonlinear reconstruction techniques (see Discussion).

A technical issue that arises in the context of metric space analysis is the infeasibility of computing the embeddings for all spike patterns across large numbers of cells (e.g. 66 cells in the data of Figure 3 produces $2^{66}$ responses). Therefore we focus on a spatially localized and overlapping population of 13 RGCs (6 ON and 7 OFF parasol cells in Figure 1B) because we can explicitly list all the $2^{13}$ possible response patterns. Training data was accrued from RGC responses to 5 repeated presentations of a 5 minute long white noise sequence. The first 4 minutes of each presentation was employed for training; the last minute was employed for testing.

We examined the similarity between the decoded stimulus and the target stimulus, for responses that, according to our learned quadratic metric, are increasingly distant from the target. Figure 4A (first column, third row) shows the spatial profile of the linearly decoded target response [1].

We next calculate the distance of this target firing pattern to all $2^{13}$ firing patterns and rank order them based on the learned metric. Figure 4A, top rows, shows firing patterns at the 1%, 2.5%, 5% and 75% percentiles. Below these firing patterns are the associated with linearly decoded stimuli, and the errors with respect to the target firing pattern. As we choose patterns farther from the target in terms of our metric, the distance between the decoded stimulus for the chosen firing pattern and target firing pattern systematically increases.

We quantify this observation in Figure 4B by randomly selecting pairs of responses from the test data and calculating the optimal linearly decoded stimuli associated with them (see Methods). We then plot the mean squared error (MSE) between the linearly decoded stimuli against the normalized metric distance between the responses. The decoding error systematically increases as the metric distance between the corresponding responses increases, for both the learned quadratic metric (blue) as well the Hamming distance (green). However, the distances generated by Hamming distance are

---

[1]Note that the reconstruction is based on the static population response pattern. We remove the time dimension by approximating ON and OFF parasol cell responses with a temporal filter with identical (but with oppositely signed) filters. Subsequent analyses are performed by only decoding the temporally filtered stimulus. The temporally filtered stimulus $\mathbf{s}$ is decoded as $\mathbf{s} = A\mathbf{r} + b$ , where parameters $A$, $b$ are estimated from RGC recordings.

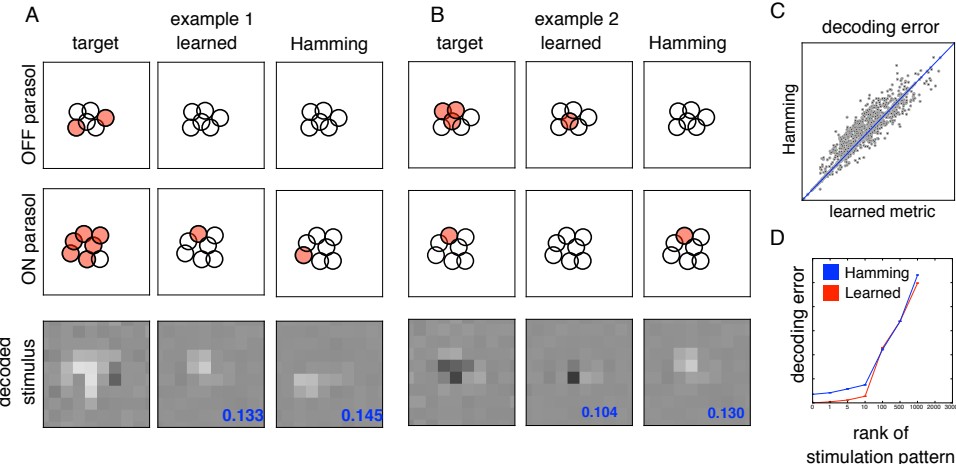

Figure 5: Learned quadratic response metric gives better stimulus reconstruction than using a Hamming metric. (A) First column: target firing pattern in ON and OFF parasol cells (shading indicates firing) and decoded stimulus. Middle column: nearest feasible, elicited firing pattern as judged by learned metric and associated decoded stimulus. The scaled mean squared error from target response is shown in the decoded figure. Right column: nearest feasible, elicited firing pattern as judged by Hamming metric and associated decoded stimulus. (B) Same as (A) but for a second target firing pattern. (C) MSE of decoded stimulus for nearest firing pattern for learned metric versus Hamming metric for all observed firing patterns. Note that $33\%, 49\%, 18\%$ of patterns reside above, on and below the line of equality, (D) Rank order of stimulation patterns selected by the learned metric (red) and Hamming metric (blue) versus MSE of decoded stimulus for the first 2000 closest patterns

discrete and therefore provide a less granular representation of the decoding errors associated with the stimuli.

### 3.4 LEARNED RESPONSE METRIC MAY IMPROVE PERFORMANCE OF RETINAL PROSTHESIS.

Using recorded experimental data, we now show how response metrics could improve the function of retinal prostheses by selecting optimal electrical stimulation patterns. For a given target response, we use the learned quadratic metric to select the best electrical stimulation pattern, and evaluate the effectiveness of this approach by linearly decoding the stimulus from the elicited responses.

Calibration of RGC responses to electrical stimulation patterns was performed by repeating a given electrical stimulation pattern 25 times, at each of 40 current levels, in a random order. Due to the limited duration of experiments, we focused on stimulation patterns in which only one electrode was active. The data was spike sorted and spiking probability was computed for each cell by averaging across trials for each electrical stimulation pattern (Mena et al., 2017). For each cell and electrode, the probability of firing as a function of stimulation current was approximated with a sigmoid function.

Since the RGC response to electrical stimulation is probabilistic, we evaluate each stimulation pattern by the *expected* distance between the elicited responses and the target firing pattern. For a quadratic response metric this can be easily computed in closed form. Given a response metric, we rank different stimulation patterns based on the expected distance to the target firing pattern. In Figure 5A and B (first columns) we show example target response patterns and the corresponding linearly decoded visual stimulus. We then analyze the best stimulation pattern determined by the learned quadratic metric, and by the Hamming distance. The responses sampled from the response distributions for the selected stimulation patterns are shown in Figures 5A and B (second and third columns each). We find that the linearly decoded stimuli were closer to the target when the stimulation was chosen via the learned response metric compared to the Hamming distance.

To quantify this behavior, we calculated the mean squared error between the decoded stimuli when the stimulation was chosen using the learned metric and the Hamming distance (Figure 5C). The

learned metric and Hamming metric identify the same stimulation pattern and hence achieve the same error for 49% for the target responses observed. However, on 33% of the target responses, the learned metric achieves lower mean squared error than the Hamming distance; conversely, the learned metric has larger MSE then Hamming distance on 18% of the target responses.

The above analysis demonstrates the benefit of using the learned metric over Hamming distance to choose the best stimulation pattern. However, the collection of available electrical stimulation patterns might change over time due to hardware or biophysical constraints. To assess the improvement in such cases, we next ask how well the learned metric performs relative to Hamming distance if we choose the $k$th best current pattern using each metric. (Figure 5D). Increasing $k$ for the learned metric leads to higher MSE in terms of the decoded stimulus. Importantly, the learned metric achieves systematically lower MSE than the Hamming distance across the nearest $k \leq 10$ stimulation patterns. These results indicate that the learned metric systematically selects better electrical stimulation patterns for eliciting reasonably close firing patterns.

## 4 DISCUSSION

The learned metric approach has two major potential implications for visual neuroscience. First, it provides a novel method to find "symbols" in the neural code of the retina that are similar in the sense that they indicate the presence of similar stimuli (Ganmor et al., 2015). Second, it has an application to retinal prosthesis technology, in which hardware constraints demand that the set of neural responses that can be generated with a device be used to effectively transmit useful visual information. For this application, a metric on responses that reflects visual stimulus similarity could be extremely useful.

The present approach differs from previously proposed spike train metrics (reviewed in (Victor, 2005)). Previous approaches have employed unsupervised techniques to cluster nearby spike patterns (Ganmor et al., 2015; Prentice et al., 2016; Gardella et al., 2017) or employed user-specified, paramteric approaches (Victor & Purpura, 1997; Aronov et al., 2003). In the case of single snapshots in time used here, the latter approach (Victor-Purpura metric) has only one degree of freedom which is a user-specified cost associated with moving spikes from one cell to another. In our proposed method, the relative importance of cell identity is learned directly from the statistics of population firing patterns.

The present work is a stepping stone towards building an encoding algorithm for retinal prostheses. In this paper, we learn the metric using light evoked responses. However, we need to estimate this metric in a blind retina, which has no light evoked responses. The convolutional metric is adaptable to any RGC population by merely noting cell types and center locations. Thus a convolutional metric could be trained on multiple healthy retinas and applied to a blind retina. Preliminary results in this direction indicate that a convolutional metric trained on half of the cells in a retinal recording (training data) generalizes to the other half (validation data), yielding performance higher than a quadratic metric (and comparable to a convolutional metric) trained directly on the validation data.

Additional techniques may also be helpful in extending our method to data involving many cells, temporal responses, and additional response structure. For example, using recurrent neural networks (Lipton et al., 2015) to embed responses may help compute distances between spiking patterns consisting of multiple time bins, perhaps of unequal length. Boosting (Freund & Schapire, 1999) may help combine multiple efficiently learned metrics for a smaller, spatially localized groups of cells. Other metrics may be developed to capture invariances learned by commonly used encoding models (Chichilnisky, 2001; Pillow et al., 2008). Also, triplet mining techniques (i.e., choosing hard negatives), a commonly used trick in computer vision, may improve efficiency (Schroff et al., 2015; Oh Song et al., 2016). Novel metrics could also be learned with additional structure in population responses, such as the highly structured correlated activity in RGCs Mastronarde (1983); Greschner et al. (2011). This noise correlation structure may be learnable using *negative* examples that destroy the noise correlations in data while preserving light response properties, by taking responses of different cells from different repeats of the stimulus.

Note that the convolutional metric outperforms the quadratic metric at both global (ROC curves) and local (precision recall curves) scales. However, using current retinal prosthesis technologies, we might be able to resolve information only up to a particular scale. For current retinal prostheses,

capturing global structure may be of greatest importance, because state-of-the-art technology has a relatively coarse vocabulary for stimulating RGCs (Humayun et al., 2012; Zrenner et al., 2011) (see also Figure 1). Specifically, the "nearest" elicited firing pattern is "far" in terms of the corresponding visual stimulus (Figure 5) . In terms of the proposed learned metric, the nearest feasible firing pattern achievable by electrical stimulation in our experiments is at the 10th percentile of all possible firing patterns. In this context, the average closest stimulation pattern, expressed as a percentile of the learned metric distances, provides a valuable benchmark to measure the performance of a prosthesis and how that performance is affected by advances in the underlying hardware and software.

## ACKNOWLEDGEMENTS

We thank Vineet Gupta for numerous helpful discussions. We thank Pawel Hottowy, Alexander Sher, Alan M. Litke, Alexandra Tikidji-Hamburyan, Georges Goetz, Nora Brackbill, Colleen Rhoades and Lauren Grosberg for help with experiments. Research funding provided by Internship program at Google Brain (NPS), DARPA Contract FA8650-16-1-765 (EJC).

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

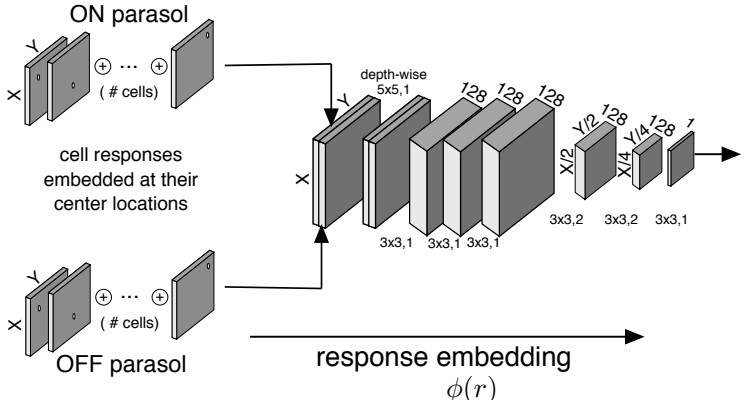

Figure 6: Convolutional network used for response embedding. The convolutional filtering is specified between each layer. For example $3 \times 3, 1$ means convolution of a $3 \times 3$ filter with stride 1.

## A  APPENDIX

### A.1  DETAILS OF THE CONVOLUTIONAL METRIC

We build a hierarchical, convolutional network to mirror the translation invariance expected in the receptive field organization of the retina. The goal of this network is to flexibly capture population activity of ON and OFF cells but employ minimal knowledge about cell receptive fields. The reason for this approach is to build a model that may be amenable to a retinal prosthetic in which the characterization of individual retinal ganglion cells is limited (Jepson et al., 2014a;b).

In particular, the network employs knowledge of the receptive field locations and firing rates of individual cells but the network is independent of the number of cells in the retina. The latter point is achieved by embedding the responses of neurons into pathways grouped by cell type. In our experiments, we focus on 2 cell types (ON and OFF parasols), thus we employ a 2 channel pathway (Kandel et al., 2000).

The network receives as input the spiking activity of ON and OFF parasols and embeds these spike patterns as one-hot vectors placed at the spatial locations of each cell's receptive field. The resulting pattern of activations is summed across all cells in the ON and OFF populations, respectively, and passed through several convolutional layers of a network. Successive layers shrink the spatial activation size of the representation, while increasing the number of filter channels (Krizhevsky et al., 2012; Simonyan & Zisserman, 2014). The final embedding response vector has 1/16th number of pixels in the stimulus and represents the flattened representation of the last layer of the network.

Let $c$ denote the number of different cells. The RGC population response is a vector $\mathbf{r} \in \{0, 1\}^c$.

- Represent responses as vectors over $\{+1, -1\}$ with $\tilde{\mathbf{r}} = 2(\mathbf{r} - 0.5)$.
- Compute the scale for each cell as a function of the mean firing rate:

$$s_i = a_0 \mu_i^3 + a_1 \mu_i^2 + a_2 \mu_i^3 + a_3 .$$

- Map each cell to its center location on a grid with spatial dimensions same as those of visual stimulus. Let $M_i$ be grid embedding on cell $i$. So, $M_i$ has zero for all positions except center of cell.
- Perform a separable $5 \times 5$ convolution of stride 1 on each $M_i$ to get RF estimate of cell, $\tilde{M}_i$.
- Add the activation of cells of the same type to get the total activation for a given cell type. Hence, activation map for each cell type $A_i = \sum_i \tilde{\mathbf{r}}_i s_i \tilde{M}_i$. Subsequent layers receive input as a two layered activation map corresponding to ON and OFF parasol cells.
- The convolutional layers further combine information accross multiple cells, of different types. The details of different layers are shown in Figure 6 and Table 1.

| | Operation | Kernel size | Stride | Feature maps | Padding | Nonlinearity |
|---|---|---|---|---|---|---|
| **Network** – $X \times Y \times 2$ input | | | | | | |
| | Separable | 5 | 1 | 1 | SAME | ReLU |
| | Convolution | 3 | 1 | 128 | SAME | ReLU |
| | Convolution | 3 | 1 | 128 | SAME | ReLU |
| | Convolution | 3 | 1 | 128 | SAME | ReLU |
| | Convolution | 3 | 2 | 128 | SAME | ReLU |
| | Convolution | 3 | 2 | 128 | SAME | ReLU |
| | Convolution | 3 | 1 | 1 | SAME | ReLU |

| | |
|---|---|
| Total number of parameters | 595229 |

| | |
|---|---|
| Normalization | Batch normalization after every convolution |
| Optimizer | Adam (Kingma & Ba, 2014) ($\alpha = 0.01$, $\beta_1 = 0.9$, $\beta_2 = 0.999$) |
| Parameter updates | 20,000 |
| Batch size | 100 |
| Weight initialization | Xavier initialization (Glorot & Bengio, 2010) |

Table 1: Details of convolutional network for embedding neural responses.

## A.2 ACCURACY OF THE LINEAR DECODER

For the latter analyses assesing the quality of metric, we reconstruct the stimulus from neural responses with linear decoding. In this section we demonstrate that even though the linear decoder is rather simplistic, the reconstructions are on-par with a non-parametric decoding method which averages the stimulus corresponding to the response pattern. In Figure 7 A, we see that the linear decoder has very similar spatial structure to the non-parametric decoder. To quantify this, we compute the mean-squared error between the two methods of decoding, normalized by the magnitude of non-parametric decoder (Figure 7 B, blue dots). The error of linear decoding is comparable to error between two non-parametric decodings computed using independent samples of stimuli (Figure 7 B, green dots). These observations show that linear decoder is a reasonable first-order approximation of encoded stimulus.

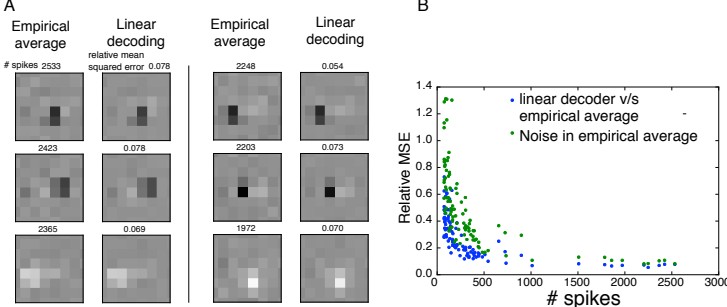

Figure 7: Accuracy of linear decoder. (A) The linearly decoded stimulus (right) and the average of stimuli (left) corresponding to 6 different most frequently occurring response patterns. The number of spikes is indicated above each non-parametric decoder (left column). The relative mean-squared error (MSE) between two decoding approaches is also indicated. (B) The relative MSE between two linear and non-parametric decoding (y-axis) v/s number of averaged spikes (x-axis) for different response patterns (blue dots). The relative MSE between non-parametric decoding using two independent set of response samples is also shown (green).

