# OpenReview forum: "Learning a neural response metric for retinal prosthesis"
_ICLR.cc/2018/Conference — Accept (Poster)_

### Official Review · AnonReviewer2 · 2017-11-22
**Really neat idea, but execution could use some work**

**Rating:** 5
**Confidence:** 4

**Review:**

The authors develop new spike train distance metrics that cluster together responses to the same stimulus, and push responses to different stimuli away from each other. Two such metrics are discussed: neural networks, and quadratic metrics. They then show that these metrics can be used to classify neural responses as coming from the same vs different stimuli, and that they outperform the naive Hamming distance metric at this task. Moreover, they show that this metric implicitly captures some structure in the neural code: more similar responses correspond to more similar visual stimuli. Finally, they discuss the implications of their metric for retinal prosthesis, and show some (fairly preliminary) data for how it could be used.

Overall, I love the concepts in this paper. I have some reasonably substantive concerns over the execution, outlined below. But I encourage the authors to consider following through on these suggestions to improve their paper: the paper's key idea is really good, and I think it's worth the effort to flesh that idea out more thoroughly.

My specific suggestions / criticisms are:

1) The quadratic metric seems only marginally better than the Hamming one (especially in Figs. 3 and 4), whereas the neural nets do much better as a metric (Fig. 3). However, most of the analyses (Figs. 4,5) use the quadratic metric. Why not use the better neural network metric for the subsequent studies of image similarity, and retinal stimulation?

2) For Figs. 4, 5, where you use linear decoders to test the stimuli corresponding to the neural responses, how good are those decoders (i.e., MSE between decoded stim and true stim.)? If the decoders are poor, then the comparisons based on those decoders might not be so meaningful. I encourage you to report the decoding error, and if it's large, to make a better decoder and use it for these studies.

3) Similarly, for Fig. 4, why not measure the MSE between the actual image frames corresponding to these neural responses? Presumably, you have the image frames corresponding to the target response, and for each of the other responses shown (i.e., the responses at different distances from the target). This would avoid any complications from sub-optimal decoders, and be a much more direct test.

(I understand that, for Fig. 5, you can't do this direct comparison, as the electrically stimulated patterns don't have corresponding image frames, so you need to decode them.)

---

### Official Review · AnonReviewer3 · 2017-11-26
**Interesting approach to learning neural response metrics**

**Rating:** 6
**Confidence:** 3

**Review:**

In their paper, the authors propose to learn a metric between neural responses by either optimizing a quadratic form or a deep neural network. The pseudometric is optimized by positing that the distance between two neural responses to two repeats of the same stimulus should be smaller than the distance between responses to different stimuli. They do so with the application of improving neural prosthesis in mind.

First of all, I am doubtful about this application: I don't think the task of neural prosthesis can ever be to produce idential output pattern to the same stimuli. Nevertheless, a good metric for neural responses that goes beyond e.g. hamming distance or squared error between spike density function would be clearly useful for understanding neural representations.

Second, I find the framework proposed by the authors interesting, but not clearly motivated from a neurobiological perspective, as the similarity between stimuli does not appear to play a role in the optimized loss function. For two similar stimuli, natural responses of neural population can be more similar than the responses to two repetitions of the same stimulus.

Third, the results presented by the authors are not convincing throughout. For example, 4B suggests that indeed the Hamming distance achieves lower error than the learned representation.

Nevertheless, it is an interesting approach that is worthwhile pursuing further.

---

### Official Review · AnonReviewer1 · 2017-11-27
**Application of metric-learning to neural population recordings in the context of prosthetics**

**Rating:** 7
**Confidence:** 4

**Review:**

* Summary of paper: The paper addresses the problem of optimizing metrics in the context of retinal prosthetics: Their goal is to learn a metric which assumes spike-patterns generated by the same stimulus to be more similar to each other than spike-patterns generated by different stimuli. They compare a conventional, quadratic metric to a neural-network based representation and a simple Hamming metric, and show that the neural-network based on achieves higher performance, but that the quadratic metric does not substantially beat the simple Hamming baseline. They subsequently evaluate the metric (unfortunately, only the quadratic metric) in two interesting applications involving electrical stimulation, with the goal of selecting stimulations which elicit spike-patterns which are maximally similar to spike-patterns evoked by particular stimuli.

* Quality: Overall, the paper is of high quality. What puzzled me, however is the fact that, in the applications using electrical stimulation in the paper (i.e. the applications targeted to retinal prosthetics, Secs 3.3 and 3.4), the authors do not actually used the well-performing neural-network based metric, but rather the quadratic metric, which is no better than the baseline Hamming metric?  It would be valuable for them to comment on what additional challenges would arise by using the neural network instead, and whether they think they could be surmonted.

* Clarity: The paper is overall clear, but specific aspects could be improved: First, it took me a while to understand (and is not entirely clear to me) what the goal of the paper is, in particular outside the setting studied by the authors (in which there is a small number of stimuli to be distinguished). Second, while the paper does not claim to provide a new metric-learning approach, it would benefit from more clearly explaining if and how their approach relates to previous approaches to metric learning.  Third, the paper, in my view, overstating some of the implications. As an example, Figure 5 is titled 'Learned quadratic response metric gives better perception than using a Hamming metric.': there is no psychophysical evaluation of perception in the paper, and even the (probably hand-picked?) examples in the figure do not look amazing.

* Originality: To the best of my knowledge, this is the first paper addressing the question of learning similarity metrics in the context of retinal prosthetics. Therefore, this specific paper and approach is certainly novel and original. From a machine-learning perspective, however, this seems like pretty standard metric learning with neural networks, and no attempt is made to either distinguish or relate their approach to prior work in this field (e.g. Chopra et al 2005, Schroff et al 2015 or Oh Song et al 2016.)

In addition, there is a host of metrics and kernels which have been proposed for measuring similarity between spike trains (Victor-Purpura) -- while they might not have been developed in the context of prosthetics, they might still be relevant to this tasks, and it would have been useful to see a comparison of how well they do relative to a Hamming metric. The paper states this as a goal ("This measure should expand upon...), but then never does that- why not?

* Significance: The general question the authors are approaching (how to improve retinal prosthetics) is,  an extremely important one both from a scientific and societal perspective. How important is the specific advance presented in this paper? The authors learn a metric for quantifying similarity between neural responses, and show that it performs better than a Hamming metric. It would be useful for the paper to comment on how they think that metric to be useful for retinal prosthetics. In a real prosthetic device, one will not be able learn a metric, as the metric learning her requires access to multiple trials of visual stimulation data, neuron-by-neuron. Clearly, any progress on the way to retinal prosthetics is important and this approach might contribute that. However, the current presentation of the manuscripts gives a somewhat misleading presentation of what has been achieved, and a more nuanced presentation would be important and appropriate.


Overall, this is a nice paper which could be of interest to  ICLR. Its strengths are that i) they identified a novel, interesting and potentially impactful problem that has not been worked on in machine learning before, ii) they provide a solution to it based on metric learning, and show that it performs better than a non-learned metrics. Its limitations are that i) no novel machine-learning methodology is used (and relationship to prior work in machine learning is not clearly described) ii) comparisons with previously proposed similarity measures of spike trains are lacking, iii) the authors do not actually use their learned, network based metric, but the metric which performs no better than the baseline in their main results, and  iv) it is not well explained how this improved metric could actually be used in the context of retinal prosthetics.

Minor comments:

  - p.2 The authors write that the element-wise product is denoted by $A \bullet B = \Tr(A^{\intercal}) B$
    This seems to be  incorrect, as the r.h.s. corresponds to a scalar.
  - p.3 What exactly is meant by “mining”?
  - p.4 It would be useful to give an example of what is meant by “similarity learning”.
  - p.4 “Please the Appendix” -> “Please see the Appendix”
  - p.5 (Fig. 3) The abbreviation “AUC” is not defined.
  - p.5 (Fig. 3B) The figure giving 'recall' should have a line indicating perfect performance, for comparison.
  - Sec 3.3: How was the decoder obtained ?
  - p.6 (Fig. 4) Would be useful to state that column below 0 is the target. Or just replace “0” by “target”.
  - p.6 (3rd paragraph) The sentence “Figure 4A bottom left shows the spatial profile of the linear decoding 20ms prior to the target response.” is unclear. It took me a very long time to realize that "bottom left" meant "column 0, 'decoded stimulus'" row. It's also unclear why the authors chose to look at 20ms prior to the target response.
  - p.6 The text says RMS distance, but the Fig. 4B caption says MSE— is this correct?

---

### Decision · Program_Chairs · 2018-01-29
**ICLR 2018 Conference Acceptance Decision**

**Decision:**

Accept (Poster)

**Comment:**

This work shows interesting potential applications of known machine learning techniques to the practical problem of how to devise a retina prosthesis that is the most perceptually useful. The paper suffers from a few methodological problems pointed out by the reviewers (e.g., not using the more powerful neural network encoding in the subsequent experiments of the paper), but is still interesting and inspiring in its current state.